# GENERALIZED INNER LOOP META-LEARNING

## ABSTRACT

Many (but not all) approaches self-qualifying as "meta-learning" in deep learning and reinforcement learning fit a common pattern of approximating the solution to a nested optimization problem. In this paper, we give a formalization of this shared pattern, which we call GIMLI, prove its general requirements, and derive a general-purpose algorithm for implementing similar approaches. Based on this analysis and algorithm, we describe a library of our design, `unnamedlib`, which we share with the community to assist and enable future research into these kinds of meta-learning approaches. We end the paper by showcasing the practical applications of this framework and library through illustrative experiments and ablation studies which they facilitate.

## 1 INTRODUCTION

Although it is by no means a new subfield of machine learning research (see e.g. Schmidhuber, 1987; Bengio, 2000; Hochreiter et al., 2001), there has recently been a surge of interest in meta-learning (e.g. Maclaurin et al., 2015; Andrychowicz et al., 2016; Finn et al., 2017). This is due to the methods meta-learning provides, amongst other things, for producing models that perform well beyond the confines of a single task, outside the constraints of a static dataset, or simply with greater data efficiency or sample complexity. Due to the wealth of options in what could be considered "meta-" to a learning problem, the term itself may have been used with some degree of underspecification. However, it turns out that many meta-learning approaches, in particular in the recent literature, follow the pattern of optimizing the "meta-parameters" of the training process by nesting one or more inner loops in an outer training loop. Such nesting enables training a model for several steps, evaluating it, calculating or approximating the gradients of that evaluation with respect to the meta-parameters, and subsequently updating these meta-parameters.

This paper makes three contributions. First, we propose a formalization of this general process, which we call **G**eneralized **I**nner Loop **M**eta-**L**earning (GIMLI), and show that it subsumes several recent approaches. The proposed formalism allows us to describe the meta-optimization process in general terms and analyse its requirements. Second, we derive a general algorithm that supports the implementation of various kinds of meta-learning fitting within the GIMLI framework and its requirements. Third, based on this analysis and algorithm, we describe a lightweight PyTorch library that enables the straightforward implementation of any meta-learning approach that fits within the GIMLI framework in canonical PyTorch, such that existing codebases require minimal changes, supporting third party module implementations and a variety of optimizers. Through a set of indicative experiments, we showcase the sort of research directions that are facilitated by our formalization and the corresponding library.

The overarching aim of this paper is not—emphatically—to purport some notion of ownership or precedence in any sense over existing efforts by virtue of having proposed a unifying formulation. Rather, in pointing out similarities under such unification, it provides theoretical and practical tools for facilitating further research in this exciting domain.

## 2 GENERALIZED INNER LOOP META-LEARNING

Whereby "meta-learning" is taken to mean the process of "learning to learn", we can describe it as a nested optimization problem according to which an outer loop optimizes meta-variables controlling the optimization of model parameters within an inner loop. The aim of the outer loop should be to

improve the meta-variables such that the inner loop produces models which are more suitable according to some criterion. In this section, we will formalize this process, which we call **G**eneralized **I**nner Loop **M**eta-**L**earning (GIMLI). We will define our terms in Section 2.1, give a formal description of the inner and outer loop's respective optimization problems in Section 2.2 and 2.3, followed by a description and proof of the requirements under which the process can be run in Section 2.4. Finally, in Section 2.5 we describe an algorithm which permits an efficient and exact implementation of GIMLI, agnostic to the model and optimizers used.

## 2.1 DEFINITIONS

Let us assume we wish to train a model parameterized by $\theta$. Let $\varphi$ describe some collection of possibly unrelated "meta-parameters" describing some aspect of the process by which we train our model. These could be, for example the learning rate or some other real-valued hyperparameters, task loss weights in multi-task learning, or perhaps the initialization of the model weights at the beginning of training. For disambiguation, we will refer to the subset of $\varphi$ describing optimization parameters by $\varphi^{opt}$, and those meta-parameters which parameterize aspects of the training loss calculation at every step (e.g. the loss itself, the task mixture, regularization terms, etc) by $\varphi^{loss}$, i.e. $\varphi = \varphi^{opt} \cup \varphi^{loss}$. We will give several concrete examples in Section 3.

Let $\mathcal{L}^{train}$ be a function of $\theta$ and $\varphi$ corresponding to the overall training objective we seek to minimize directly or indirectly (e.g. via an upper bound, a Monte Carlo estimate of an expectation, etc). Other elements of the objective such as the dataset, and hyperparameters or other aspects of training not covered by $\varphi$, are assumed to be implicit and held fixed in $\mathcal{L}^{train}$ for notational simplicity. With this, the process by which we train a model to obtain test-time parameters $\theta^*$ can be formalized as shown in Equation 1[1].

$$\theta^* = \mathbf{argmin}(\theta; \mathcal{L}^{train}(\theta, \varphi)) \tag{1}$$

We assume that this search will be performed using an iterative gradient-based method such as some form of gradient descent as formalized in Section 2.2, and thus that the derivative $\nabla_\theta \mathcal{L}^{train}(\theta, \varphi)$ exists, as an obvious requirement.

Furthermore, let $\mathcal{L}^{val}$ be a function exclusively of $\theta$ (except the specific case where $\varphi$ describes some aspect of the model, e.g. initialization, as is done in MAML (Finn et al., 2017)) describing some objective we wish to measure the post-training performance of the model against. This could be a validation set from the same task the model was trained on, from a collection of other tasks, or some other measure of the quality of the model parameterized by $\theta^*$. Typically, the aforementioned iterative process is not run to convergence, but rather to the point where an intermediate set of model parameters is considered satisfactory according to some criterion (e.g. best model under $\mathcal{L}^{val}$ for a fixed budget of training steps, or after no improvement is seen for a certain number of training steps, etc.), and these intermediate parameters will serve as $\theta^*$.

## 2.2 TRAINING

The process by which we typically approximate $\theta^*$ against $\mathcal{L}^{train}$ decomposes into a sequence of updates of $\theta$. At timestep $t$, we compute $\theta_{t+1}$ from $\theta_t$ by first computing a step-specific loss $\ell_t^{train}(\theta_t, \varphi^{loss})$. Note that this loss may itself also be a function of step-specific factors, such as the training data used for that timestep, which we leave implicit here for notational simplicity. Following this, we typically compute or approximate (e.g. as in reinforcement learning) the gradients $\nabla_{\theta_t} \ell_t^{train}(\theta_t, \varphi^{loss})$ of this loss with regard to $\theta_t$, by using algorithms such as backpropagation (Rumelhart et al., 1985). We then typically use an optimizer to "apply" these gradients to the parameters $\theta_t$ to obtain updated parameters $\theta_{t+1}$. As such, optimization processes such as e.g. Adagrad (Duchi et al., 2011) may be stateful, in that they exploit gradient history to produce adaptive "local" learning rates. Because some aspects of the optimization process may be covered by our

---

[1]Throughout this paper, we will write $\arg\min_x$ in functional form as follows to facilitate formalizing nested optimization, with $x$ a variable and $E$ an expression (presumably a function of $x$):

$$\mathbf{argmin}(x; E) := \arg\min_x E$$

choice of $\varphi$, we denote this optimization step at time-step $t$ by $\mathbf{opt}_t$ as shown in Equation 2, where timestep-specific attributes of the optimizer are left implicit by our use of the step subscript.

$$\theta_{t+1} = \mathbf{opt}_t(\theta_t, \varphi^{opt}, G_t) \quad \text{where} \quad G_t = \nabla_{\theta_t} \ell_t^{train}(\theta_t, \varphi^{loss}) \tag{2}$$

For example, where we might use SGD as an optimizer, with the learning-rate being a meta-variable $\varphi^{opt}$, we could instantiate equation 2 as follows:

$$\mathbf{opt}_t(\theta_t, \varphi^{opt}, G_t) := \theta_t - \varphi^{opt} \cdot G_t \quad \text{where} \quad G_t = \nabla_{\theta_t} \ell_t^{train}(\theta_t, \varphi^{loss})$$

The estimation of $\theta^*$ from Equation 1 using $T + 1$ training updates according to Equation 2 yields a double recurrence in both $\theta_t$ and $G_t$ (as it is a function of $\theta_t$, as outlined in Equation 3).

$$\theta^* \approx \mathbf{opt}_T(\theta_T, \varphi^{opt}, G_T) = \mathbf{opt}_T(P_T, \varphi^{opt}, \nabla_{\theta_T} \ell_T^{train}(P_T, \varphi^{loss}))$$

$$\text{where } P_T = \mathbf{opt}_{T-1}(\theta_{T-1}, \varphi^{opt}, G_{T-1}) = \mathbf{opt}_{T-1}(P_{T-1}, \varphi^{opt}, \nabla_{\theta_{T-1}} \ell_{T-1}^{train}(P_{T-1}, \varphi^{loss}))$$

$$\vdots$$

$$\text{where } P_1 = \mathbf{opt}_0(\theta_0, \varphi^{opt}, G_0) = \mathbf{opt}_0(\theta_0, \varphi^{opt}, \nabla_{\theta_0} \ell_0^{train}(\theta_0, \varphi^{loss})) \tag{3}$$

From this we see that the optimization process used to train a variety of model types, with a variety of optimization methods, can be described as a function yielding test-time model parameters $\theta^*$ potentially as a function of parameter history $\theta_1, \ldots, \theta_T$ and of meta-parameters $\varphi$ (if any exist). While this may seem like a fairly trivial formalization of a ubiquitous training process, we will see in Section 2.4 that if a few key requirements are met, this process can be nested as the inner loop within an outer loop containing—amongst other things—a meta-training process. From this process, described in Section 2.3, we estimate values of $\varphi$ which improve our training process against some external metric.

## 2.3 META-TRAINING

We now describe the outer loop optimization problem which wraps around the inner loop described in Section 2.2. Through this process, we seek a value $\varphi^*$ which ensures that the training process $\mathcal{L}^{train}(\theta, \varphi^*)$ produces parameters $\theta^*$ which perform best against some metric $\mathcal{L}^{val}(\theta^*)$ which we care about. The formalization and decomposition of this "meta-training process" into nested optimization problems is shown in Equation 4.

$$\varphi^* = \mathbf{argmin}\left(\varphi; \mathcal{L}^{val}(\theta^*)\right)$$

$$= \mathbf{argmin}\left(\varphi; \mathcal{L}^{val}\left(\mathbf{argmin}\left(\theta; \mathcal{L}^{train}(\theta, \varphi)\right)\right)\right) \tag{4}$$

In this section, we introduce a formalization of an iterative process allowing us to approximate this nested optimization process. Furthermore, we describe a general iterative algorithm by which the process in Equation 4 can be approximated by gradient-based methods while jointly estimating $\theta^*$ according to the process in Equation 1.

An iterative process by which we can estimate $\varphi^*$ given Equation 4 following the sort of decomposition of the training process described Equation 1 into the training process described in Equations 2–3 is described below. Essentially, an estimate of $\theta^*$ is obtained following $T + 1$ steps training as outlined in Equation 3, which is then evaluated against $\mathcal{L}^{val}$. The gradient of this evaluation, $\nabla_\varphi \mathcal{L}^{val}(\theta^*)$ is then obtained through backpropagation, and used to update $\varphi$. Using $\tau$ as time-step counter to symbolize the time-scale being different from that used in the "inner loop", we formalize this update in Equation 5 where $\mathbf{metaopt}_\tau$ denotes the optimization process used to update $\varphi_\tau$ using $M_\tau$ at that time-step.

$$\varphi_{\tau+1} = \mathbf{metaopt}_\tau(\varphi_\tau, M_\tau) \quad \text{where} \quad M_\tau = \nabla_{\varphi_\tau} \mathcal{L}^{val}(\theta^*) \tag{5}$$

$$= \mathbf{metaopt}_\tau(\varphi_\tau, \nabla_{\varphi_\tau} \mathcal{L}^{val}(\mathbf{opt}_T(P_T, \varphi^{opt}, G_T)))$$

$$\text{where } P_T = \mathbf{opt}_{T-1}(P_{T-1}, \varphi^{opt}, \nabla_{\theta_{T-1}} \ell_{T-1}^{train}(P_{T-1}, \varphi^{loss}))$$

$$\vdots$$

$$\text{where } P_1 = \mathbf{opt}_0(\theta_0, \varphi^{opt}, G_0)$$

## 2.4 KEY REQUIREMENTS

For gradient-based meta-learning as formalized in Section 2.3—in particular through the process formalized in Equation 5—to be possible, a few key requirements must be met. We enumerate them here, and then discuss the conditions under which they are met, either analytically or through choice of model, loss function, or optimizer.

I. $\mathcal{L}^{val}$ is a differentiable function of its input, i.e. the derivative $\nabla_{\theta^*}\mathcal{L}^{val}(\theta^*)$ exists.

II. The optimization process **opt** in Equation 2 is a differentiable[2] function of $\theta$ and $G$[3].

III. Either or both of the following conditions hold:

    (a) there exist continuous optimization hyperparameters (e.g. a learning rate $\alpha$) covered by $\varphi^{opt}$ (e.g. $\alpha \subseteq \varphi^{opt}$) and **opt** in Equation 2 is a differentiable function of $\varphi^{opt}$, *or*

    (b) the gradient $G_t$ for one or more time-steps in Equations 1–3 is a function of $\varphi^{loss}$ (i.e. the derivative $\nabla_{\varphi^{loss}}\ell^{train}(\theta, \varphi^{loss})$ exists).

In the remainder of this section, we show that based on the assumptions outlined in Section 2.1, namely the existence of gradients on the validation objective $\mathcal{L}^{val}$ with regard to model parameters $\theta$ and of gradients on the training objective $\mathcal{L}^{train}$ with regard to both model parameters $\theta$ and meta-parameters $\varphi$, there exist gradients on the validation objective $\mathcal{L}^{val}$ with regard to $\varphi$. We will then, in the next section, demonstrate how to implement this process by specifying an update algorithm for meta-variables.

Implementing an iterative process as described in Equation 5 will exploit the chain rule for partial derivatives in order to run backpropagation. The structure of the recurrence means we need to ensure that $\nabla_{\theta_t}\mathcal{L}^{val}(\theta^*)$ exists for $t \in \{0, \ldots, T\}$ in order to compute, for all such $\theta_t$, gradient paths $(\nabla_{\theta_t}\mathcal{L}^{val}(\theta^*)) \cdot (\nabla_{\varphi}\theta_t)$. We can prove this exists in virtue of the chain rule for partial derivatives and the requirements above:

1. By the chain rule, $\nabla_{\theta_t}\mathcal{L}^{val}(\theta_*) = (\nabla_{\theta^*}\mathcal{L}^{val}(\theta_*)) \cdot (\nabla_{\theta_t}\theta^*)$ exists if $\nabla_{\theta^*}\mathcal{L}^{val}(\theta_*)$ exists (and it does, by Requirement I) and $\nabla_{\theta_t}\theta^* = \nabla_{\theta_t}\theta_{T+1}$ exists.

2. *Idem*, $\nabla_{\theta_t}\theta_{T+1} = (\nabla_{\theta_{t+1}}\theta_{T+1}) \cdot (\nabla_{\theta_t}\theta_{t+1})$ exists by recursion over $t$ if for all $i \in [1, T]$, $\nabla_{\theta_i}\theta_{i+1}$ exists, which is precisely what Requirement II guarantees.

Therefore $\nabla_{\theta_t}\mathcal{L}^{val}(\theta^*)$ exists for $t \in \{0, \ldots, T\}$, leaving us to demonstrate that $\nabla_{\varphi}\theta_t$ is defined for all relevant values of $t$ as a consequence of requirements I–III. We separately consider the case of $\varphi^{loss}$ and $\varphi^{opt}$ as defined in Section 2.1:

1. For $\nabla_{\varphi^{opt}}\theta_t$, the gradients trivially exist as a consequence of Requirement IIIa.

2. For $\nabla_{\varphi^{loss}}\theta_t$, by the chain rule, $\nabla_{\varphi^{loss}}\theta_t = (\nabla_{G_{t-1}}\theta_t) \cdot (\nabla_{\varphi^{loss}}G_{t-1})$. From Requirement II, it follows that $\nabla_{G_{t-1}}\theta_t$ exists, and from Requirement IIIb, it follows that $\nabla_{\varphi^{loss}}G_{t-1}$ exists, therefore so does $\nabla_{\varphi^{loss}}\theta_t$.

Putting these both together, and having covered the union of $\varphi^{opt}$ and $\varphi^{loss}$ by exhaustion, we have shown that the gradients $\nabla_{\varphi}\mathcal{L}^{val}(\theta^*)$ can be obtained by composition over the gradient paths $(\nabla_{\theta_t}\mathcal{L}^{val}(\theta^*))(\nabla_{\varphi}\theta_t)$ for all $t \in [1, T]$. In Section 2.5 we show how to implement the exact and efficient calculation of $\nabla_{\varphi}\mathcal{L}^{val}(\theta^*)$. To complete this section, we indicate the conditions under which requirements I–III hold in practice.

Requirement I is simply a function of the choice of evaluation metric used to evaluate the model after training as part of $\mathcal{L}^{val}$. If this is not a differentiable function of $\theta^*$, e.g. BLEU (Papineni et al., 2002) in machine translation, then a proxy metric can be selected for meta-training (e.g. negative

---

[2]For stateful optimizers, we assume that, where the state is itself a function of previous values of the network parameters (e.g. moving averages of parameter weights), it is included in the computation of gradients of the meta-loss with regard to those parameters.

[3]This requirement is typically only satisfied by deterministic optimizers. However, for stochastic optimizers, if there exists a reparameterization trick (see Kingma & Welling, 2013) for the distribution from which gradients or other components of the update are sampled, the requirement may also be satisfied.

log-likelihood of held out data), or gradient estimation methods such as REINFORCE (Williams, 1992) can be used.

Requirement II is a function of the choice of optimizer, but is satisfied for most popular optimizers. We directly prove that this requirement holds for SGD (Robbins & Monro, 1951) and for ADA-GRAD (Duchi et al., 2011) in Appendix A, and prove it by construction for a wider class of common optimizers in the implementation and tests of the software library described in Section 4.

Requirement IIIa is a function of the choice of hyperparameters and optimizer, but is satisfied for at least the learning rate in most popular optimizers. Requirement IIIb is a function of the choice of loss function $\ell^{train}$ (or class thereof), in that $\nabla_{\theta_t} \ell_t^{train}(\theta_t, \varphi^{loss})$ needs to exist and be a differentiable function of $\varphi$. Usually, this requirement is held where $\varphi$ is a multiplicative modifier of $\theta_t$. For algorithms such as Model Agnostic Meta-Learning (Finn et al., 2017, MAML), this requirement is equivalent to saying that the Hessian of the loss function with regard to the parameters exists.

## 2.5 THE GIMLI UPDATE ALGORITHM

In Algorithm 1, we present the algorithm which permits the computation of updates to meta-variables through the nested iterative optimization process laid out above. To clearly disentangle different gradient paths, we employ the mathematical fiction that is the "stop-gradient" operator, which is defined as an operator which maintains the truth of the following expression:

$$\underset{\mathbf{stop}(x)}{[\![-]\!]} : \quad \forall f \left[ \left( \underset{\mathbf{stop}(x)}{[\![f(x)]\!]} = f(x) \right) \wedge \left( \underset{\mathbf{stop}(x)}{\nabla_x [\![f(x)]\!]} = 0 \right) \right]$$

As will be shown below, this will allow us to decompose the computation of updates to the meta-variables through an arbitrarily complex training process, agnostic to the models and optimizers used (subject to the requirements of Section 2.4 being satisfied), into a series of local updates passing gradient with regard to the loss back through the inner loop steps. This is akin to backpropagation through time (BPTT; Rumelhart et al., 1985), a method which has been adapted to other nested optimization processes with various constraints or restrictions (Andrychowicz et al., 2016; Franceschi et al., 2017; 2018).

---

**Algorithm 1** The GIMLI update loop

---

**Require:** Current model parameters $\theta_t$, meta-parameters $\varphi_\tau$
**Require:** Number of meta-parameter updates $I$, length of unrolled inner loop $J$
1: **for** $i = 0$ **to** $I - 1$ **do**
2:     Segment meta-parameters $\varphi_i^{opt}, \varphi_i^{loss} \leftarrow \mathbf{split}(\varphi_{\tau+i})$
3:     Copy model state $\theta_0' \leftarrow \theta_t$, optimizer state $\mathbf{opt}_0' \leftarrow \mathbf{opt}_t$
4:     **for** $j = 0$ **to** $J - 1$ **do**
5:         Compute inner gradient $G_j \leftarrow \nabla_{\theta_j'} \ell_{t+j}^{train}(\theta_j', \varphi_i^{loss})$ and retain gradient graph state
6:         Update inner model $\theta_{j+1}' \leftarrow \mathbf{opt}_j'(\theta_j', \varphi_i^{opt}, G_j)$
7:     **end for**
8:     Initialize accumulators: $A_i^{opt} \leftarrow \mathbf{zerosLike}(\varphi_i^{opt}); A_i^{loss} \leftarrow \mathbf{zerosLike}(\varphi_i^{loss})$
9:     Compute $B_J \leftarrow \nabla_{\theta_J'} \mathcal{L}^{val}(\theta_J')$
10:     **for** $j' = J - 1$ **to** $0$ **do**
11:         Compute optimizer-gradient derivative $O_{j'} \leftarrow \nabla_{G_{j'}} \mathbf{opt}_{j'}'(\theta_{j'}', \varphi_i^{opt}, G_{j'})$
12:         Update $A_i^{opt} \leftarrow A_i^{opt} + B_{j'+1} \cdot (\nabla_{\varphi_i^{opt}} \mathbf{opt}_{j'}'( \underset{\mathbf{stop}(\varphi_i^{opt})}{[\![\theta_{j'}']\!]} , \varphi_i^{opt}, \underset{\mathbf{stop}(\varphi_i^{opt})}{[\![G_{j'}]\!]} ))$
13:         Update $A_i^{loss} \leftarrow A_i^{loss} + B_{j'+1} \cdot O_{j'} \cdot (\nabla_{\varphi_i^{loss}} \nabla_{\theta_{j'}'} \ell_{t+j'}^{train}( \underset{\mathbf{stop}(\varphi_i^{loss})}{[\![\theta_{j'}']\!]} , \varphi_i^{loss}))$
14:         Compute $B_{j'} \leftarrow B_{j'+1} \cdot ((\nabla_{\theta_{j'}} \mathbf{opt}_{j'}'(\theta_{j'}', \varphi_i^{opt}, \underset{\mathbf{stop}(\theta_{j'})}{[\![G_{j'}]\!]} )) + O_{j'} \cdot (\nabla_{\theta_{j'}'}^2 \ell_{t+j'}^{train}(\theta_{j'}', \varphi_i^{loss})))$
15:     **end for**
16:     Update meta-parameters $\varphi_{\tau+i+1} \leftarrow \mathbf{metaopt}(\varphi_{\tau+i}, \mathbf{join}(A_i^{opt}, A_i^{loss}))$
17: **end for**
18: **return** Updated meta-parameters $\varphi_{\tau+I}$

---

Each iteration through the loop defined by lines 1–17 does one gradient-based update of meta-parameters $\varphi$ using the optimizer employed in line 16. Each such iteration, we first (line 3) copy

the model and optimizer state (generally updated through some outer loop within which this update loop sits). We then (lines 4–7) compute a series of $J$ updates on a copy of the model, preserving the intermediate gradient computations $G_0, \ldots, G_{J-1}$, intermediate model parameters $\theta'_0, \ldots, \theta'_J$ (sometimes confusingly referred to as "fast weights", following Hinton & Plaut (1987), within the meta-learning literature), and all associated activations. These will be reused in the second stage (lines 10–15) to backpropagate higher-order gradients of the meta-loss, computed on line 9 through the optimization process that was run in lines 4–7. In particular, in lines 12 and 13, local (i.e. time step-specific) gradient calculations compute part of the gradient of $\nabla_{\varphi_i^{opt}} \mathcal{L}^{val}(\theta'_J)$ and $\nabla_{\varphi_i^{loss}} \mathcal{L}^{val}(\theta'_J)$, which is stored in accumulators which contain the exact respective gradients by the end of loop. What permits this efficient local computation is the dynamic programming calculation of the partial derivative $B_{j'} = \nabla_{\theta'_{j'}}$ as function of only $B_{j'+1}$ and timestep-specific gradients, implementing a second-order variant of BPTT through reverse-mode differentiation.

## 3 EXAMPLES AND RELATED WORK

In this section, we highlight some instances of meta-learning which are instances of GIMLI, before discussing related approaches involving support for nested optimization, with applications to similar problems. The aim is not to provide a comprehensive literature review, which space would not permit. Rather, in pointing out similarity under our GIMLI formulation, we aim to showcase that rich and diverse research has been done using this class of approaches, where yet more progress indubitably remains to be made. This is, we believe, the strongest motivation for the development of libraries such as the one we present in Section 4 to support the implementation of algorithms that fall under the general algorithm derived in Section 2.

### 3.1 EXAMPLES

Many of the papers referenced below contain excellent and thorough reviews of the literature most related to the type of meta-learning they approach. In the interest of brevity, we will not attempt such a review here, but rather focus on giving examples of a few forms of meta-learning that fit the GIMLI framework (and thus are supported by the library presented in Section 4), and briefly explain why.

One popular meta-learning problem is that of learning to optimize hyperparameters through gradient-based methods (Bengio, 2000; Maclaurin et al., 2015; Luketina et al., 2016; Franceschi et al., 2017), as an alternative to grid/random search (Bergstra & Bengio, 2012) or Bayesian Optimization (Mockus et al., 1978; Pelikan et al., 1999; Bergstra et al., 2011; Snoek et al., 2012). Here, select continuous-valued hyperparameters are meta-optimized against a meta-objective, subject to the differentiability of the optimization step, and, where relevant, the loss function. This corresponds to Requirements II and IIIa of Section 2.4 being met, i.e. GIMLI being run with select optimizer hyperparameters as part of $\varphi^{opt}$. To give a simple concrete example, in the approaches of Bengio (2000) and Maclaurin et al. (2015), the only meta-variable is the learning rate $\alpha$, i.e. $\varphi = \varphi^{opt} = \alpha$.

A related problem is that of learning the optimizer wholesale as a parametric model (Hochreiter et al., 2001; Andrychowicz et al., 2016; Duan et al., 2016), typically based on recurrent architectures. Again, here the optimizer's own parameters are the optimizer hyperparameters, and constitute the entirety of $\varphi^{opt}$ as used within GIMLI. Requirements II and IIIa are trivially met through the precondition that such optimizers models have parameters with regard to which their output (and losses that are a function thereof) is differentiable. As a concrete example, in the work of Andrychowicz et al. (2016), an RNN with parameters $\phi$ is meta-learned, and models the updates made to parameters during training. In our formalism, this would correspond to setting as meta-variable the parameters of this update network, i.e. $\varphi^{opt} = \phi$.

More recently, meta-learning approaches such as MAML (Finn et al., 2017; Antoniou et al., 2018) and its variants/extensions have sought to use higher-order gradients to meta-learn model/policy initializations in few-shot learning settings. In GIMLI, this corresponds to setting $\theta_0 = \varphi^{loss}$, which then is not an explicit function of $\ell^{train}$ in Equation 3, but rather is implicitly its argument through updates to the inner model over the unrolled optimization. All requirements in Section 2.4 must be satisfied (save IIIa, with Requirement IIIb further entailing that $\ell^{train}$ be defined such that the second derivative of the function with regard to $\theta$ exists (i.e. is non-zero).

Finally, recent work by Chebotar et al. (2019) has introduced the $ML^3$ framework for learning unconstrained loss functions as parametric models, through exploiting second-order gradients of a meta-loss with regard to the parameters of the inner loss. This corresponds, in GIMLI, to learning a parametric model of the loss parameterized by $\varphi^{loss}$.

## 3.2 RELATED WORK

In a sense, many if not all of the approaches discussed in Section 3.1 qualify as "related work", but here we will briefly discuss approaches to the general problem of formalizing and supporting implementations of problems that fit within the nested optimization specified by GIMLI.

The first is work by Franceschi et al. (2018) which describes how several meta-learning and hyper-parameter optimization approaches can be cast as a bi-level optimization process, akin to our own formalization in 2.3. This fascinating and relevant work is highly complementary to the formalization and discussion presented in our paper. Whereas we focus on the requirements according to which gradient-based solutions to approaches based on nested optimization problems can be found in order to drive the development of a library which permits such approaches to be easily and scalably implemented, their work focuses on analysis of the conditions under which exact gradient-based solutions to bi-level optimization processes can be approximated, and what convergence guarantees exist for such guarantees. In this sense, this is more relevant to those who wish to analyze and extend alternatives to first-order approximations of algorithms such as MAML, e.g. see work of Nichol & Schulman (2018) or Rajeswaran et al. (2019).

On the software front, the library `learn2learn` (Arnold et al., 2019) addresses similar problems to that which we will present in Section 4. This library focuses primarily on providing implementations of existing meta-learning algorithms and their training loops that can be extended with new models. In contrast, the library we present in Section 4 is "closer to the metal", aiming to support the development of new meta-learning algorithms fitting the GIMLI definitions with as little resort to non-canonical PyTorch as possible. A recent parallel effort, Torchmeta (Deleu et al., 2019) also provides a library aiming to assist the implementation of meta-learning algorithms, supplying useful data-loaders for meta-training. However, unlike our approach described in 4, it requires re-implementation of models using their functional/stateless building blocks, and for users to reimplement the optimizers in a differentiable manner.

## 4 THE `unnamedlib` LIBRARY

In this section, we provide a high-level description of the design and capabilities of `unnamedlib`,[4] a PyTorch (Paszke et al., 2017) library aimed at enabling implementations of GIMLI with as little reliance on non-vanilla PyTorch as possible. In this section, we first discuss the obstacles that would prevent us from implementing this in popular deep learning frameworks, how we overcame these in PyTorch to implement GIMLI. Additional features, helper functions, and other considerations when using/extending the library are provided in its documentation.

## 4.1 OBSTACLES

Many deep learning frameworks offer the technical functionality required to implement GIMLI, namely the ability to take gradients of gradients. However, there are two aspects of how we implement and train parametric models in such frameworks which inhibit our ability to flexibly implement Algorithm 1.

The first obstacle is that models are typically implemented statefully (e.g. `torch.nn` in PyTorch, `keras.layers` in Keras (Chollet et al., 2015), etc.), meaning that the model's parameters are encapsulated in the model implementation, and are implicitly relied upon during the forward pass. Therefore while such models can be considered as functions theoretically, they are not pure functions practically, as their output is not uniquely determined by their explicit input, and equivalently the parameters for a particular forward pass typically cannot be trivially overridden or supplied at call

---

[4]We will publicly release the library described in this section in the coming weeks under the Apache license. References to the library name have been anonymized throughout the paper to preserve the double-blind reviewing process.

time. This prevents us from tracking and backpropagating over the successive values of the model parameters $\theta$ within the inner loop described by Equation 3, through an implicit or explicit graph.

The second issue is that, as discussed at the end of Section 2.4 and in Appendix A, even though the operations used within popular optimizers are mathematically differentiable functions of the parameters, gradients, and select hyperparameters, these operations are not tracked in various framework's implicit or explicit graph/gradient tape implementations when an optimization step is run. This is with good reason: updating model parameters in-place is memory efficient, as typically there is no need to keep references to the previous version of parameters. Ignoring the gradient dependency formed by allowing backpropagation through an optimization step essentially makes it safe to release memory allocated to historical parameters and intermediate model states once an update has been completed. Together, these obstacles essentially prevent us from practically satisfying Requirement II of Section 2.4.

## 4.2 MAKING STATEFUL MODULES STATELESS

As we wish to track and backpropagate through intermediate states of parameters during the inner loop, we keep a record of such states which can be referenced during the backward pass stage of the outer loop in Algorithm 1. The typical way this is done in implementations of meta-learning algorithms such as MAML is to rewrite a "stateless" version of the inner loop's model, permitting the use, in each invocation of the model's forward pass, of weights which are otherwise tracked on the gradient graph/tape. While this addresses the issue, it is an onerous and limiting solution, as exploring new models within such algorithms invariably requires their reimplementation in a stateless style. This typically prevents the researcher from experimenting with third-party codebases, complicated models, or those which requiring loading pre-trained weights, without addressing a significant and unwelcome engineering challenge.

A more generic solution, permitting the use of existing stateful modules (including with pre-loaded activations), agnostic to the complexity or origin of the code which defines them, is to modify the run-time instance of the model's parent class to render them effectively function, a technique often referred to as "monkey-patching". The high-level function `unnamedlib.monkeypatch()` does this by taking as argument a `torch.nn.Module` instance and the structure of its nested sub-modules. As it traverses this structure, it clones the parent classes of submodule instances, leaving their functionality intact save for that of the `forward` method which implements the forward pass. Here, it replaces the call to the forward method with one which first replaces the stateful parameters of the submodule with ones provided as additional arguments to the patched `forward`, before calling the original class's bound `forward` method, which will now used the parameters provided at call time. This method is generic and derived from first-principles analysis of the `torch.nn.Module` implementation, ensuring that any first or third-party implementation of parametric models which are subclasses of `torch.nn.Module` and do not abuse the parent class at runtime will be supported by this recursive patching process.

## 4.3 MAKING OPTIMIZERS DIFFERENTIABLE

Again, as part of our need to make the optimization process differentiable in order to satisfy Requirement II of Section 2.4, the typical solution is to write a version of SGD which does not modify parameters in-place, but treated as a differentiable function of the input parameters and hyperparameters akin to any other module in the training process. While this, again, is often considered a satisfactory solution in the meta-learning literature due to its simplicity, it too is limiting. Not only does the inability to experiment with other inner loop optimizers prevent research into the applicability of meta-learning algorithms to other optimization processes, the restriction to SGD also means that existing state-of-the-art methods used in practical domains cannot be extended using meta-learning methods such as those described in Section 3, lest they perform competitively when trained with SGD.

Here, while less generic, the solution provided by the high-level function `unnamedlib.get_diff_optim()` is to render a PyTorch optimizer instance differentiable by mapping its parent class to a differentiable reimplementation of the instance's parent class. The reimplementation is typically a copy of the optimizer's step logic, with in-place operations being replaced with gradient-tracking ones (a process which is syntactically simple to execute in PyTorch).

To this, we add wrapper code which copies the optimizer's state, and allows safe branching off of it, to permit "unrolling" of the optimization process within an inner loop (cf. the recurrence from Equation 3) without modifying the initial state of the optimizer (e.g. to permit several such unrolls, or to preserve state if inner loop optimizer is used elsewhere in the outer loop). Most of the optimizers in `torch.optim` are covered by this method. Here too, a runtime modification of the parent optimizer class could possibly be employed as was done for `torch.nn.Modules`, but this would involve modifying Python objects at a far finer level of granularity. We find that supporting a wide and varied class of optimizers is a sufficient compromise to enable further research.[5]

## 5  EXPERIMENTS

In this section, we briefly present some results of experiments using select existing methods from Section 3.1, to show case how `unnamedlib` can be used to simply implement ablation studies and searches over model architectures, optimizers, and other aspects of an experiment. This could, naturally, be done without appeal to the library. However, in such cases, changing the model architecture or optimizer requires re-implementing the model functionally or optimization step differentiably. Here such changes require writing no new code (excluding the line where the model is defined).

### 5.1  META-LEARNING LEARNING RATES WITH HIGHER GRANULARITY

As the first application for `unnamedlib`, we set up a simple meta-learning task where we meta-optimize the learning rate. Employing a handcrafted annealing schedule for learning rates has been the *de facto* approach to improving a learning algorithm. While scheduled annealing can help to boost the final performance, it usually requires significant hand-tuning of the schedule. In contrast, we adjust learning rates automatically using meta-learning, which `unnamedlib` enables for arbitary models and optimizers.

We take an image classification model `DenseNet-BC(k=12)` (Huang et al., 2016), that provides competitive state-of-the-art results on CIFAR10, and modify its training procedure by replacing the multi-step annealing schedule of learning rate with a meta-optimizer. Specifically, we treat learning rate for each inner optimizer's parameter group as a separate meta-parameter that we will meta-learn. Doing so allows individual model parameters to have finer learning rates, that are adjusted automatically. The GIMLI algorithm provides for a clean implementation of such training procedure.

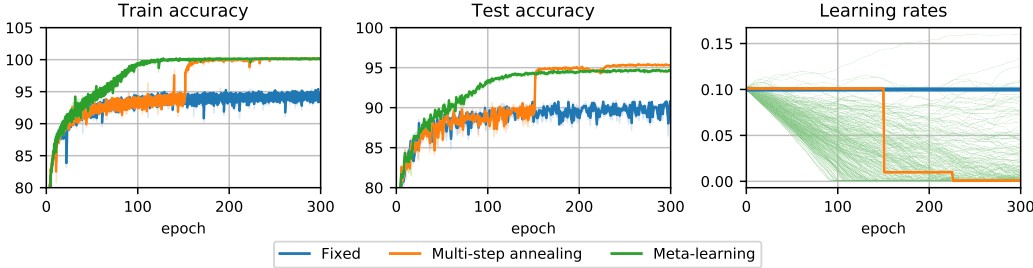

Figure 1: Comparison of meta-learned learning rates against fixed and multi-step annealed for training DenseNet-BC(k=12) on CIFAR10. We observe convergence near state-of-the-art with better sample complexity that using a hand-designed annealing schedule.

We first train two baseline variants of `DenseNet-BC(k=12)` using setup from (Huang et al., 2016). In the first variant we keep the learning rate fixed to 0.1 for all 300 epochs, while the second configuration takes advantage of a manually designed multi-step annealing schedule, which drops learning rate by 10 after 150 and 225 epochs. For the meta-learning variant, we split the training set into two disjoint pieces, one for training and another for validation, in proportion of 99 : 1. We then use per parameter group learning rates (299 in total) for meta-optimization, initializing each learning rate to 0.1. We perform one meta-update step after each epoch, where we unroll inner optimizer for

---

[5] We also provide documentation as to how to write differentiable third party optimizers and supply helper functions such as `unnamedlib.register_optim()` to register them for use with the library at runtime.

Table 1: Results from ablating MAML++ architecture and inner optimizers. "Our base" results are from our VGG model trained with SGD in our MAML++ variant, and "our best" results show the best test accuracy found, with the best model/optimizer combination shown in the text below.

| | omniglot test accuracy | | | | miniImageNet test accuracy | |
| | 5 Way | | 20 Way | | 5 Way | |
| Approach | 1 Shot | 5 Shot | 1 Shot | 5 Shot | 1 Shot | 5 Shot |
|---|---|---|---|---|---|---|
| MAML++ (Antoniou et al., 2018) | $99.53 \pm 0.26\%$ | $99.93 \pm 0.09\%$ | $97.65 \pm 0.05\%$ | $99.33 \pm 0.03\%$ | $52.15 \pm 0.26\%$ | $68.32 \pm 0.44$ |
| MAML++ (Our base) | $99.62 \pm 0.08\%$ | $99.86 \pm 0.02\%$ | $97.21 \pm 0.11\%$ | $99.13 \pm 0.13\%$ | $56.33 \pm 0.27\%$ | $75.13 \pm 0.67\%$ |
| MAML++ (Our best) | $99.91 \pm 0.05\%$ | $99.87 \pm 0.03\%$ | $99.00 \pm 0.33\%$ | $99.76 \pm 0.01\%$ | $56.33 \pm 0.27\%$ | $76.73 \pm 0.52\%$ |
| | resnet-4+SGD | resnet-4+SGD | resnet-12+SGD | resnet-8+SGD | vgg+SGD | resnet-8+SGD |

15 steps using batches from the training set, and compute meta-test error on the validation set over 10 batches from the validation set. We use batch size of 16 for meta-update, rather than 64 as in the base training loop. We use Adam (Kingma & Ba, 2014) with default parameters as a choice for meta-optimizer. We average results over 3 random seeds. Figure 1 demonstrates that our method is able reach the state-of-the-art performance faster.

## 5.2 ABLATING MAML'S MODEL ARCHITECTURE AND INNER OPTIMIZER

The `unnamedlib` library enables the exploration of new MAML-like models and inner-loop optimizers, which historically has required non-trivial implementations of the fast-weights for the model parameters and inner optimizers as done in Antoniou et al. (2018); Deleu et al. (2019). These ablations can be important for squeezing the last few bits of accuracy on well-established tasks and baselines that are already near-optimal as shown in Chen et al. (2019), and is even more important for developing new approaches and tasks that deal with different kinds of data and require adaptation to be done over non-standard operations.

To illustrate how easy `unnamedlib` makes these ablations, in this section we take a closer look at different model architecture and optimizer choices for the MAML++ approach (Antoniou et al., 2018). MAML++ uses a VGG network with a SGD inner optimizer for the the Omniglot (Lake et al., 2015) and Mini-Imagenet (Vinyals et al., 2016; Ravi & Larochelle, 2016) tasks. We start with the official MAML++ code and evaluation procedure and use `unnamedlib` to ablate across VGG, ResNet, and DenseNet models and SGD and Adam optimizers. We provide more details about these in Appendix B. We denote the combination of model and inner optimizer choice with `<model>+<opt>`. One finding of standalone interest is that we have kept most of the features from MAML++, *except* we significantly increase the base `VGG+SGD` performance by using batch normalization in training mode everywhere as in (Finn et al., 2017) instead of using per-timestep statistics and parameters as MAML++ proposes. In theory, this enables more inner optimization steps to be rolled out at test time, which otherwise is not possible with MAML++ because of the per-timestep information, for simplicity in this paper we have not explored this and keep every algorithm, optimizer, and mode to five inner optimization steps. When using Adam as the inner optimizer, we initialize the first and second moment statistics to the statistics from the outer optimizer, which is also Adam, and learn per-parameter-group learning rates and rolling average $\beta$ coefficients. Our results in Table 1 show that we are able to push the accuracy of MAML++ slightly up with this ablation. We note that this pushes the performance of MAML++ closer to that of state-of-the-art methods such as LEO (Rusu et al., 2018). Appendix B shows our full experimental results, and we note that in some cases Adam slightly outperforms SGD for a particular model.

## 6 CONCLUSION

To summarize, we have presented GIMLI, a general formulation of a wide class of existing and potential meta-learning approaches, listed and proved the requirements that must be satisfied for such approaches to be possible, and specified a general algorithmic formulation of such approaches. We've described a lightweight library, `unnamedlib`, which extends PyTorch to enable the easy and natural implementation of such meta-learning approaches at scale. Finally we've demonstrated some of its potential applications. We hope to have made the case not only for the use of the mathematical and software tools we present here, but have also provided suitable encouragement for other researchers to use them and explore the boundaries of what can be done within this broad class of meta-learning approaches.

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

# A   OPTIMIZER DIFFERENTIABILITY

## A.1   SGD

$$\nabla_\varphi \theta_{t+1} = \nabla_\varphi \mathbf{SGD}(\theta_t, G_t) = \nabla_\varphi[\theta_t - \alpha G_t] = \nabla_\varphi \theta_t - G_t \nabla_\varphi \alpha - \alpha \nabla_\varphi G_t$$

where $\alpha$ is the learning rate, and $\nabla_\varphi \alpha$ is defined iff $\alpha \subseteq \varphi$ else $G_t \nabla_\varphi \alpha = 0$

$$\nabla_{\theta_t} \theta_{t+1} = \nabla_{\theta_t} \mathbf{SGD}(\theta_t, G_t) = \nabla_{\theta_t}[\theta_t - \alpha G_t] = \nabla_{\theta_t} \theta_t - \alpha \nabla_{\theta_t} G_t = 1 - \alpha \nabla_{\theta_t}^2 \ell_t^{train}(\theta_t, \varphi)$$

## A.2   ADAGRAD

$$\nabla_\varphi \theta_{t+1} = \nabla_\varphi \mathbf{Adagrad}(\theta_t, G_t) = \nabla_\varphi[\theta_t - \frac{\eta}{\sqrt{\sum_{i=1}^t G_i^2}} G_t]$$

$$= \nabla_\varphi \theta_t - \frac{\eta \nabla_\varphi G_t}{\sqrt{\sum_{i=1}^t G_i^2}} - \frac{G_t \nabla_\varphi \eta}{\sqrt{\sum_{i=1}^t G_i^2}} + \frac{\eta G_t \sum_{i=1}^t G_i \nabla_\varphi G_i}{\left(\sum_{i=1}^t G_i^2\right)^{\frac{3}{2}}}$$

where $\eta$ is the global learning rate, and $\nabla_\varphi \eta$ is defined iff $\eta \subseteq \varphi$ else $\dfrac{G_t \nabla_\varphi \eta}{\sqrt{\sum_{i=1}^t G_i^2}} = 0$

$$\nabla_{\theta_t} \theta_{t+1} = \nabla_{\theta_t} \mathbf{Adagrad}(\theta_t, G_t) = \nabla_{\theta_t}[\theta_t - \frac{\eta}{\sqrt{\sum_{i=1}^t G_i^2}} G_t]$$

$$= \nabla_{\theta_t} \theta_t - \frac{\eta \nabla_{\theta_t} G_t}{\sqrt{\sum_{i=1}^t G_i^2}} + \frac{\eta G_t^2 \nabla_{\theta_t} G_t}{\left(\sum_{i=1}^t G_i^2\right)^{\frac{3}{2}}}$$

$$= 1 - \frac{\eta \nabla_{\theta_t}^2 \ell_t^{train}(\theta_t, \varphi)}{\sqrt{\sum_{i=1}^t G_i^2}} + \frac{\eta G_t^2 \nabla_{\theta_t}^2 \ell_t^{train}(\theta_t, \varphi)}{\left(\sum_{i=1}^t G_i^2\right)^{\frac{3}{2}}}$$

## B    MAML++ EXPERIMENTS: ADDITIONAL INFORMATION

Table 2 shows all of the architectures and optimizers we ablated. The table is missing some rows as we only report the results that successfully completed running three seeds within three days on our cluster.

We use the following model architectures, which closely resemble the vanilla PyTorch examples for these architectures but are modified to be smaller for the few-shot classification setting:

- `vgg` is the VGG architecture (Simonyan & Zisserman, 2014) variant used in MAML++ (Antoniou et al., 2018)

- `resnet-N` is the ResNet architecture (He et al., 2016). `resnet-4` corresponds to the four blocks having just a single layer, and `resnet-8` and `resnet-12` have respectively 2 and 3 layers in each block

- `densenet-8` is the DenseNet architecture (Huang et al., 2017) with 2 layers in each block

We follow TADAM (Oreshkin et al., 2018) and do not use the initial convolutional projection layer common in the full-size variants of the ResNet and DenseNet.

We can also visualize how the learning rates and rolling momentum terms (with Adam) change over time when they are being learned. We show this in Figure 2 and Figure 3 for the 20-way 1-shot Omniglot experiment with the `resnet-4` architecture, which is especially interesting as Adam outperforms SGD in this case. We find that most of the SGD learning rates are decreased to near-zero except for a few select few that seem especially important for the adaptation. Adam exhibits similar behavior with the learning rates where most except for a select for are zeroed for the adaptation, and the rolling moment coefficients are particularly interesting where the first moment coefficient $\beta_1$ becomes relatively evenly spread throughout the space while $\beta_2$ splits many parameter groups between low and high regions of the space.

Table 2: Full MAML++ model and inner optimizer sweep search results.

| dataset | nway | kshot | model | inner_optim | mean acc | std |
|---|---|---|---|---|---|---|
| mini_imagenet_full_size | 5 | 1 | densenet-8 | SGD | 46.08 | 1.40 |
| | | | resnet-12 | SGD | 51.06 | 1.51 |
| | | | resnet-4 | Adam | 49.71 | 3.71 |
| | | | | SGD | 54.36 | 0.23 |
| | | | resnet-8 | SGD | 54.16 | 1.35 |
| | | | vgg | Adam | 47.93 | 11.64 |
| | | | | SGD | 56.33 | 0.27 |
| | | 5 | densenet-8 | SGD | 65.29 | 0.98 |
| | | | resnet-12 | Adam | 37.40 | 3.64 |
| | | | | SGD | 69.14 | 3.19 |
| | | | resnet-4 | Adam | 76.33 | 0.71 |
| | | | | SGD | 74.48 | 0.77 |
| | | | resnet-8 | Adam | 68.03 | 15.19 |
| | | | | SGD | 76.73 | 0.52 |
| | | | vgg | Adam | 72.82 | 2.36 |
| | | | | SGD | 75.13 | 0.67 |
| omniglot_dataset | 5 | 1 | densenet-8 | SGD | 99.54 | 0.33 |
| | | | resnet-4 | SGD | 99.91 | 0.05 |
| | | | vgg | Adam | 99.62 | 0.08 |
| | | | | SGD | 99.62 | 0.08 |
| | | 5 | densenet-8 | SGD | 99.86 | 0.05 |
| | | | resnet-4 | SGD | 99.87 | 0.03 |
| | | | vgg | Adam | 99.86 | 0.04 |
| | | | | SGD | 99.86 | 0.02 |
| | 20 | 1 | densenet-8 | SGD | 93.20 | 0.32 |
| | | | resnet-12 | SGD | 99.00 | 0.33 |
| | | | resnet-4 | Adam | 98.31 | 0.09 |
| | | | | SGD | 96.31 | 0.15 |
| | | | resnet-8 | SGD | 98.50 | 0.15 |
| | | | vgg | Adam | 96.15 | 0.16 |
| | | | | SGD | 97.21 | 0.11 |
| | | 5 | densenet-8 | SGD | 97.24 | 0.26 |
| | | | resnet-12 | SGD | 99.69 | 0.17 |
| | | | resnet-4 | Adam | 99.44 | 0.23 |
| | | | | SGD | 99.71 | 0.03 |
| | | | resnet-8 | SGD | 99.76 | 0.01 |
| | | | vgg | Adam | 98.74 | 0.04 |
| | | | | SGD | 99.13 | 0.13 |

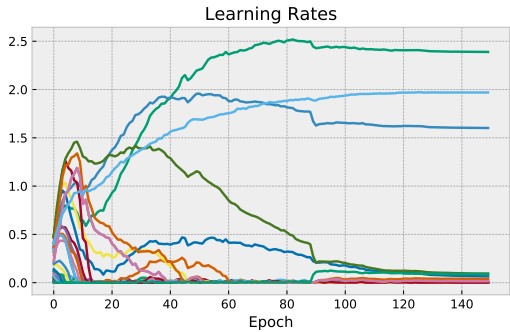

Figure 2: The learning rates during a training run of a VGG network with SGD as the inner optimizer for 20-way 1-shot mini-imagenet classification. The colors show the parameter groups within the model.

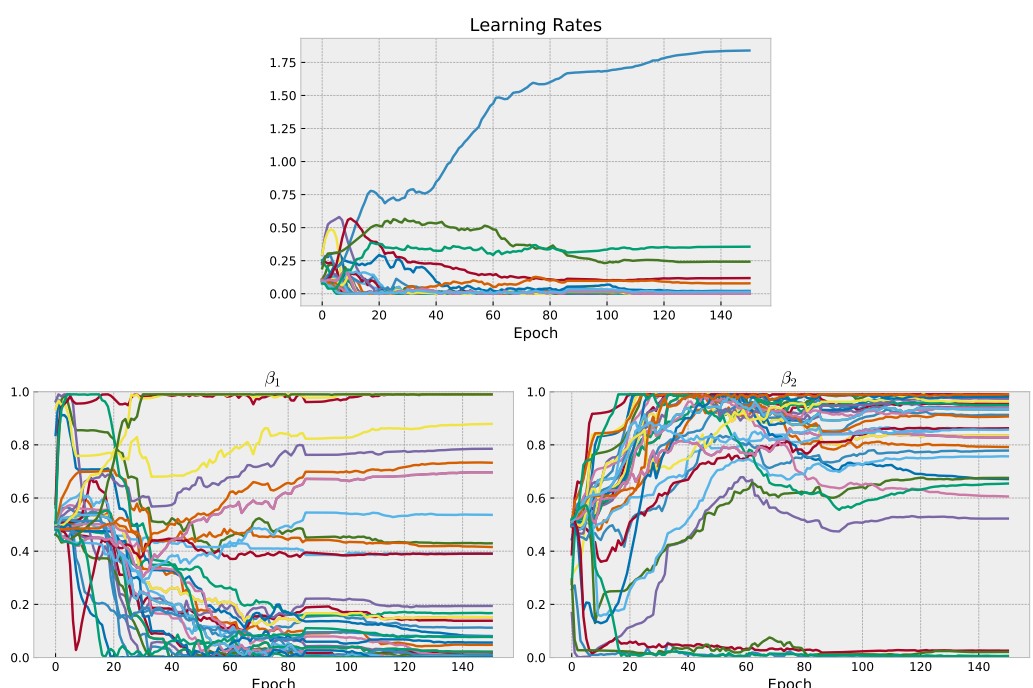

Figure 3: The learning rates during a training run of a VGG network with Adam as the inner optimizer for 20-way 1-shot mini-imagenet classification. The colors show the parameter groups within the model.

