# OpenReview forum: "Generalized Inner Loop Meta-Learning"
_ICLR.cc/2020/Conference — Reject_

### Official Review · AnonReviewer3 · 2019-10-21
**Official Blind Review #3**

**Rating:** 3

**Review:**

The authors propose the general formulation of recent meta-learning methods and propose a good library to use.

Pros:
1. The general formulation of recent meta-learning methods is reasonable.
2. The proposed library is easy to use.

Cons:

The paper lacks technical novelty. I understand the goal of this paper is to build a library. However, the paper only describes a general formulation for recent meta-learning methods (e.g., MAML) and implement the formulation. It is better to clarify and some key engineering challenges and do the corresponding experiments.

In addition, in the experiment parts, the authors only compare the results with MAML++. It will be more convincing if the authors can analyze other popular meta-learning methods (e.g.. Prototypical network [1], meta-LSTM [2]).

Another suggestion is that the authors can give some examples to connect current meta-learning models with the proposed general formulation. For example, the meaning of \phi_i^opt, \phi_i^loss in MAML, Prototype, Reptile, etc.

It is better to explain the meaning of different colors in Figure 3.

[1] Snell, Jake, Kevin Swersky, and Richard Zemel. "Prototypical networks for few-shot learning." Advances in Neural Information Processing Systems. 2017.
[2] Ravi, Sachin, and Hugo Larochelle. "Optimization as a model for few-shot learning." ICLR (2016).



Decision after rebuttal: I have read the authors' responses. Like review 2, I also think the "generalization" is overclaimed, it only provides a general formulation. Thus, I finally decide to keep my score.

**Experience Assessment:**

I have published one or two papers in this area.

**Review Assessment: Checking Correctness Of Derivations And Theory:**

N/A

**Review Assessment: Checking Correctness Of Experiments:**

I carefully checked the experiments.

**Review Assessment: Thoroughness In Paper Reading:**

I read the paper thoroughly.

---

> ### Author Response · Authors · 2019-11-10
> **Reply to Reviewer 3**
>
> We thank the reviewer for their review and suggestions. We are happy that they found the formalization clear and the library easy to use. We note that the main (if not only?) argument in favour of rejection seems to be “lack of technical novelty”. With all due respect, we believe the reviewer’s assessment is based on a misunderstanding both of the contributions of the paper, and of the ICLR CFP. We will address both matters here before replying to the additional comments and suggestions.
>
> First, the goal of the paper is not specifically to build a library. The purpose of the library presented in section 4 is entirely complementary to the material presented in sections 2-3. The main deep learning frameworks provide engineering obstacles to the easy implementation of algorithms fitting the framework of section 2, including all those exemplified in section 3 (we encourage the reviewer to look at implementations thereof if they are not already authors of such implementations, or haven’t seen them, to see what we mean). The purpose of the anonymized library is to make it easy to implement new and existing methods fitting the patterns discussed in sections 2-3 by addressing the engineering challenges presented at the beginning of section 4. In this respect, it is an entirely separate contribution.
>
> Second, noting that the CFP for ICLR calls for papers describing, amongst other things, “implementation issues, parallelization, software platforms, hardware”, we hope you will feel that even as a stand-alone contribution it is relevant and acceptable material for an ICLR publication.
>
> Furthermore, the two contributions in section 2 are orthogonal to the engineering contribution addressed in section 4. Our value proposition for section 2 is first that we formalize the general pattern of a variety of approaches and prove the (often implicit) requirements for such approaches to be possible, and second that we describe the general form of an algorithm covering a variety of existing (and hopefully future) approaches, some of which are discussed in section 3.
>
> In summary, we dispute that there is not both a significant technical contribution in the engineering challenges in section 4 (which are explicitly within the scope of the call for papers), and in the complementary formalization, proofs, and algorithm from section 2. We would appreciate if the reviewer could further detail their thinking regarding this. If they agree, we would appreciate it if they reconsidered their assessment and score.
>
> We now turn to the specific comments.
>
> > It will be more convincing if the authors can analyze other popular meta-learning methods (...)
>
> We can certainly do this in further work, but in what way would it be “more convincing” (especially since the work the reviewer refers to is significantly older)? The state of the art certainly has moved on since MAML++ (presented at ICLR’19 a mere 6 months ago), but the point of section 5.2 is emphatically not to produce SOTA results, but rather to showcase the sort of experimental research that is enabled and facilitated by the library presented in section 4: namely that by removing the need to reimplement models and optimizers from scratch when implementing MAML-style methods, proper ablation studies can be run, and they show improvements over reported results for MAML++ (as might be expected for other methods such as those proposed by the reviewer). If this isn’t a positive contribution to the community, who will be able to freely use the library to develop and experiment with new methods, then we would be grateful if the reviewer could explain in what way it is insufficient.
>
> > Another suggestion is that the authors can give some examples to connect current meta-learning models with the proposed general formulation (...)
>
> Please note that we do this in section 3 already (in the paragraph describing MAML), stating that \varphi^loss = \theta_0 (i.e. the meta-variable is the initial state of the model at the beginning of a task). We try to offer similar indications for other related work in this section. If there are any where you feel this needs expanding, please flag, and we will happily expand this section. Please note we have written the paper under the more strict 8 page limit, so details like this can be added while maintaining a paper length in line with most other submissions.
>
> Regarding methods such as reptile, or first order MAML, they do not change what the meta-variable is, but rather how the process of backpropagating through the unrolled optimization can be approximated. This is somewhat out of the scope of this paper, but if you feel this omission warrants a footnote to clarify this point, please let us know.
>
> > It is better to explain the meaning of different colors in Figure 3.
>
> We agree. They are different parameter groups, but we will add detail here. Thank you for taking the time to read the appendix! :)

---

> ### Author Response · Authors · 2019-11-14
> **Paper update changelog**
>
> Dear Reviewer,
>
> We have made modifications to the manuscript based on our discussion. For your convenience, please find, below, a log of the changes made that pertain to our discussion. Please note that the paper has still been kept under the page limit as a result of these changes.
>
> We’ve updated the manuscript to add specific examples of how select existing approaches fit the Gimli formalism in Section 3. Thank you for your suggestion.
>
> We have clarified the purpose of the different colours in the figures at the end of the appendix.
>
> Thanks again for your excellent suggestions.

---

### Official Review · AnonReviewer1 · 2019-10-24
**Official Blind Review #1**

**Rating:** 3

**Review:**

This work presented a general formulation of a wide class of existing meta-learning approaches, and proved the requirements that must be satisfied for such approaches to be possible.

Half of the work is focused on describing the unnamedlib library, which extends PyTorch to enable the easy
and natural implementation of such meta-learning approaches.

The early sections are interesting, especially section 2, which gives some great insights to the existing inner loop pattern in meta-learning. However, from section 3, the paper has turned to examples and related works, where I was hoping the author would give more detailed analysis of the pattern. My concern is the authors have spent too much space on the unnamedlib library. So http://www.jmlr.org/mloss/ might be a more suitable place for publication.

**Experience Assessment:**

I have published one or two papers in this area.

**Review Assessment: Checking Correctness Of Derivations And Theory:**

I did not assess the derivations or theory.

**Review Assessment: Checking Correctness Of Experiments:**

I assessed the sensibility of the experiments.

**Review Assessment: Thoroughness In Paper Reading:**

I read the paper at least twice and used my best judgement in assessing the paper.

---

> ### Author Response · Authors · 2019-11-10
> **Reply to Reviewer 1**
>
> Thank you for your review. We are happy to hear you found the paper interesting, especially with regard to the formalization in section 2, which forms roughly half of the content of the paper on its own. One of the main contributions of the paper is, in fact, providing the mathematical tooling to describe existing and future work within this framework. This offers several benefits, such as providing provable requirements on the optimizers, losses, and space of models for this sort of meta-learning to be possible, providing a common notation for describing new approaches, and providing a formalism within which to compare and contrast existing approaches.  We believe is, in itself, a meaningful enough contribution to justify publication, and we hope you agree.
>
> Regarding the criticisms, we understand that there are two aspects that the reviewer thinks can be improved: the first is that the related work in section 3 could give more concrete examples of how the patterns in section 2 apply; the second is that “too much space on the unnamedlib library” and another venue might be more appropriate.
>
> Regarding the first point, we agree with the reviewer that analyzing the mathematical patterns on the chosen examples in Section 3 would add value and are happy to address this. Would it be satisfactory to the reviewer if we were to address this by revising the paper during the review period, to the point where they would consider revising their assessment?
>
> Regarding the question of appropriateness of section 4 and the amount of detail therein, we respectfully disagree. The ICLR 2020 CFP specifically calls for, amongst other topics, papers discussing “implementation issues, parallelization, software platforms, hardware”. We think section 4 uncontroversially fits this particular bullet point, as it describes implementation issues facing the class of meta-learning approaches fitting the formalism from section 2, and describes how we overcame them in a library (software platform) we are releasing to the public. Would the reviewer be prepared to revise their assessment in light of this, or at least give us further indication as to why this aspect of the work is nonetheless unsuitable for ICLR?

---

> ### Author Response · Authors · 2019-11-14
> **Paper update changelog**
>
> Dear Reviewer,
>
> We have made modifications to the manuscript based on our discussion. For your convenience, please find, below, a log of the changes made that pertain to our discussion. Please note that the paper has still been kept under the page limit as a result of these changes.
>
> We’ve updated the manuscript to add specific examples of how select existing approaches fit the Gimli formalism in Section 3. Thank you for your suggestion.

---

### Official Review · AnonReviewer2 · 2019-10-27
**Official Blind Review #2**

**Rating:** 3

**Review:**

Summary:
The authors present a PyTorch based framework for performing second-order
reverse mode autodiff for meta-learning.

First, the authors present a formalization of a general prototypical
meta-learning setting.
They then provide an algorithm that solves this problem via gradient based
optimization.
Finally, perhaps the main contribution is a specific PyTorch implementation
of said algorithm.

The type of meta-learning setting the authors consider is one where a gradient
based inner loop optimizer finds $\theta^\star$ by performing a finite number of steps.
The inner loop optimizer is parameterized through $\varphi$ that consists of
two parts $\varphi^\text{loss}$ and $\varphi^\text{opt}$.
The parameters $\varphi^\text{loss}$ are somehow part of the loss used for
training in the inner loop. Example: Regularization paramter.
The parameters $\varphi^\text{opt}$ do not occur in the loss but in the
optimizer step. Example: Learning rate.

Example of an inner loop step:
$\theta^{k+1} := \theta^k - \alpha (\nabla L(\theta^k) + \lambda \nabla R(\theta^k))$
where $\theta$ are the parameters of a neural network, $L$ is the training loss,
$R$ is the regularizer, $\alpha$ is the learning rate, $\lambda$ is the regularization parameter.
In this example we would have $\varphi = (\alpha \lambda)^T$.

The authos assume $\theta^K$, the output of the inner loop after $K$ steps to
be differentiable wrt $\varphi$.
Furthermore, the meta-learning loss is assumed to be differentiable wrt $\theta$
so that a gradient of the meta-learning loss wrt to $\varphi$ can be computed.
The authors also assume the meta-learning loss to be sufficient smooth in $\varphi$
such that a gradient based optimization can even be used for meta learning to
a local optimum.

The authors explicitly write down the reverse mode auto differentiation of
the inner loop and show how to, in that way, compute the gradient of the
meta-learning loss wrt to $\varphi$.

The reverse (adjoint) mode auto differentiation of the above example inner loop step
is the following step (iterated over in reverse down from $k = K$ to $k = 1$:
$\bar \theta^k := (I - \alpha (\nabla^2 L(\theta^k) + \lambda \nabla^2 R(\theta^k)))^T \bar \theta^{k+1}$
$\bar \alpha := \bar \alpha - (\nabla l(\theta^k) + \lambda \nabla R(\theta^k))^T \bar \theta^{k+1}$
$\bar \lambda := \bar \lambda - \alpha \nabla R(\theta^k)^T \bar \theta^{k+1}$
where $\bar \alpha$ accumulates the gradient of $\theta$ wrt $\alpha$ and
$\bar \lambda$ accumulates the gradient of $\theta$ wrt $\lambda$.

The authors give some implementation details specific to some frameworks necessary
for implementing such "gradient of an inner loop".

The authors present experiments where they show how to meta-learn learning rates
with their framework.
They also how their framework can be used to quickly implement a MAML type
meta-learning optimizer ablation study comparing various combinations of
architecture, optimizer and inner loop steps etc..

Recommendation:
I propose to reject the paper.
In my eyes the only contribution is the implementation of a meta-learner in
PyTorch based on well known methods.
The provided unifying formalization is theoretically inaccurate (see below)
and to me come across as merely a motivation for their framework
(but no value added compared to existing literature).
There is no new insight provided on the software engineering level either as
far as I can see.

Detailed comments:
- Page 1: ...provides tooling for analysing the provable requirements...
	seems like a complicated way of saying something that could be said simple

- Page 2: Without loss of generality, ...
	You are assuming a parametric model. Not sure what generality this phrase
	refers to.

- Page 2: A formalization as $\theta^\star = argmin(\theta, L(\theta, \varphi))$
	is inaccurate in the sense that it does not acknowledge the existing of
	multiple optima.
	I would recommend not to use the $argmin$ operator here, since $argmin$ for
	something like a neural network would for example either return a global
	optimum (which no optimizer used in practice finds, and is not ment here)
	or would take on a set value with multiple local minima for example.

	In the same context, the authors should mention the issues about uniqueness
	of optima (we are not even really finding optima when training neural networks),
	implicit functions / implicit differentiation

	In the context of using stochastic optimizers one should also at least
	mention something about the differentiability of outputs of such optimizers
	and how they potentially depend on randomness of mini-batches
	(what if different randomness is used with the same or a perturbed
	hyper parameter?)

- Page 3: You mention the potential statefulness of the optimizer.
	Why not explicitly carry it in the math notation?
	Probably things would get cluttered but saying it should be covered within
	$\varphi$ does not seem reasonable to me.

- Page 3: While this may seem like a fairly trivial formalization...
	Yes, but also nesting this in an outer loop is fairly trivial in the sense
	that it is a well known approach.

- Page 4: there exist continuous hyperparam...
	If they are not continuous then they should not even occur in this
	formalization so saying there exist... does not make much sense to me here

	$\alpha \subseteq \varphi^\text{opt}$ implies that $\varphi^\text{opt}$ is
	a set from notation although we are treating it as a vector everywhere else

- Page 4: All of section 2.4 seems somewhat trivial to me, but I guess that is
	highly subjective.

- Page 5: in the definition of stop operator perhaps use $:\Leftrightarrow$

- Page 5: Perhaps explicitly mention how your approach differs from a reverse
	mode differentiation of training or if it does not differ, say this.

- Page 14: When talking about _S_GD (instead of just GD) perhaps mention
	something about non-existence of mini-batch randomness / being deterministic


**Experience Assessment:**

I have read many papers in this area.

**Review Assessment: Checking Correctness Of Derivations And Theory:**

I assessed the sensibility of the derivations and theory.

**Review Assessment: Checking Correctness Of Experiments:**

I assessed the sensibility of the experiments.

**Review Assessment: Thoroughness In Paper Reading:**

I made a quick assessment of this paper.

---

> ### Author Response · Authors · 2019-11-10
> **Reply to Reviewer 2 (part 1)**
>
> We thank the reviewer for their detailed comments. We appreciate the effort that has been put into this review, but believe there to be some fairly important misunderstandings (in addition to some very helpful comments) which we wish to discuss with the reviewer.
>
> The reviewer makes four arguments in favour of rejection:
>
> The only contribution is an implementation of the algorithm in part 2 in pytorch.
> No new engineering insights in section 4.
> The provided unifying formalization is theoretically inaccurate.
> The provided unifying formalization does not add value compared to related literature, and just motivates the implementation.
>
> We will address these four points below, in the hope that the reviewer is willing to engage in discussion with us regarding these issues to help improve the paper, or where relevant, willing to revise their assessment if they agree there has been a misunderstanding.
>
> Regarding point 1, with all due respect, this is simply not the case. To remind the reviewer of the main contributions of this paper: First, we propose a formalization of the process of optimizing the training process through gradient-based methods, and show that it subsumes several recent approaches. Second, we derive a general algorithm that supports the implementation of various kinds of meta-learning fitting this framework. Third, we describe (and release) a lightweight PyTorch library that enables the straightforward implementation of any meta-learning approach that fits within this framework.
>
> The purpose of the library presented in section 4 is entirely complementary to the material presented in sections 2-3. The main deep learning frameworks provide engineering obstacles to the easy implementation of algorithms fitting the framework of section 2, including all those exemplified in section 3 (we encourage the reviewer to look at implementations thereof if they are not already authors of such implementations, or haven’t seen them, to see what we mean). The purpose of the anonymized library is to make it easy to implement new and existing methods fitting the patterns discussed in sections 2-3 by addressing the engineering challenges presented at the beginning of section 4. In this respect, the library is an entirely separate contribution. Additionally, noting that the CFP for ICLR calls for papers describing, amongst other things, “implementation issues, parallelization, software platforms, hardware”, we hope you will feel that even as a stand alone contribution it is relevant and acceptable material for an ICLR publication.
>
> Regarding point 2, we are not sure how to respond, and hope the reviewer will expand on their point given our rebuttal. We genuinely do address a key engineering problem, in particular in section 4.2. If this is not sufficiently novel, can you please point us to related work (including repos) which address this challenge, and permit the off-the-shelf usage of existing third-party pre-trained model for MAML, reverse-mode differentiation, or related methods? To our knowledge, there are none, and in this respect, an engineering challenge and roadblock for the community is addressed in this work.
> Regarding point 3, we believe that are some misunderstandings, which we attempt to address in more detail below. We will address the finer points of discussion below, and hope this assuages the reviewer’s concerns, but if not, could the outstanding issues causing theoretical inaccuracy be flagged so the paper can be rectified?
>
> Finally, regarding point 4, we first refer back to our rebuttal to point 1: the two contributions in section 2 (a formalization of the process of optimizing the training process through gradient-based methods, and a general algorithm that supports the implementation of various kinds of meta-learning fitting this framework) are orthogonal to the engineering contribution addressed in section 4 (a discussion of engineering roadblocks, and of how we solved them in unnamedlib). Our value proposition for section 2 is first that we formalize the general pattern of a variety of approaches and prove the (often implicit) requirements for such approaches to be possible, and second that we describe the general form of an algorithm covering a variety of existing (and hopefully future) approaches, some of which are discussed in section 3. We claim to the reviewer that this adds value: we are unaware of existing work treating on a comprehensive discussion of the requirements for a wide class of meta-learning methods. We are unaware of equivalently general algorithmic formulation which covers a variety of applications, to guide their implementation. If the reviewer knows of such, we would appreciate pointers to work not covered in our related work section. If not, is the reviewer willing to reconsider their judgement in light of this?

---

> > ### Author Response · Authors · 2019-11-10
> > **Reply to Reviewer 2 (part 2)**
> >
> > We now turn to the detailed comments:
> >
> > > - Page 1: ...provides tooling for analysing the provable requirements...
> > > seems like a complicated way of saying something that could be said simple
> >
> > We are always happy to simplify. Would replacing “The proposed formalism provides tooling for analysing the provable requirements of the meta-optimization process, and allows for describing the process in general terms.” with “The proposed formalism allows us to describe the meta-optimization process in general terms and analyse its requirements.”?
> >
> > > - Page 2: Without loss of generality, ...
> > > You are assuming a parametric model. Not sure what generality this phrase refers to.
> >
> > We see what you mean, we meant without loss of generality within the space of parametric models, but this sentence evidently has caused confusion. We will cut “without loss of generality”.
> >
> > > - Page 2: A formalization as theta* = argmin(theta, L(theta, varphi)) is inaccurate in the sense that it does not acknowledge the existing of multiple optima. I would recommend not to use the  operator here, since  for something like a neural network would for example either return a global optimum (which no optimizer used in practice finds, and is not ment here) or would take on a set value with multiple local minima for example.
> > > In the same context, the authors should mention the issues about uniqueness of optima (we are not even really finding optima when training neural networks), implicit functions / implicit differentiation
> > I think there is some misunderstanding here, which perhaps we can avoid for other readers by suitably clarifying the text. argmin(theta, L(theta, varphi)) represents the optimization problem we are trying to solve when we train a parametric model. This is agnostic to whether a minimum exists (although we typically pick loss functions for which it does) or whether it is unique. Our framework works just as well when in the ideal case, argmin returns a set of equivalent optima as we would in practice just select an arbitrary element. Furthermore, we do not assume that we are actually solving this problem in practice, but rather estimating theta* through an iterative gradient-based method (which of course is susceptible to local minima, etc) and has no guarantee of converging on a global minimum. We believed that by stating that we estimate theta*, at the beginning of Section 2.2, it was made clear that equation 1 is a description of the training process under idealised conditions, as the rest of the paper deals with meta-learning to optimize the iterative optimization process (not the idealized one), but we can add a sentence or two clarifying the intent of equation 1 in section 2.1. In particular, we can say “approximate” instead of “estimate”. Would that sufficiently clarify matters, given the above?
> >
> > > In the context of using stochastic optimizers one should also at least mention something about the differentiability of outputs of such optimizers and how they potentially depend on randomness of mini-batches (what if different randomness is used with the same or a perturbed hyper parameter?)
> >
> > Most optimizers are, given data sampled from the data distribution and model parameters, deterministic (e.g. all optimizers in core pytorch except lbfgs). You are right that for stochastic optimizers, things are a little different, although that is technically catered to by requirement II of section 2.4. We can add an explicit mention of this here if you think it will make things clearer, or in Section 2.4. What do you think would be more suitable, given your reading?
> >
> > > - Page 3: You mention the potential statefulness of the optimizer.
> > > Why not explicitly carry it in the math notation? Probably things would get cluttered but saying it should be covered within varphi does not seem reasonable to me.
> >
> > The state of the optimizers is not covered within varphi^{opt}. We leave it implicit by adding a time index for notational simplicity, following the convention used for other things like loss functions throughout the paper. We are happy to add state explicitly as something like opt(theta_t, varphi^{opt}_t, G_t, S_t) instead of opt_t(theta_t, varphi^{opt}_t, G_t) if you think something is gained, but we are not sure what that adds, and it would just further clutter the equations. If you still think this would clarify ambiguity somehow, please let us know and we will amend.

---

> > > ### Author Response · Authors · 2019-11-10
> > > **Reply to Reviewer 2 (part 3)**
> > >
> > > > - Page 3: While this may seem like a fairly trivial formalization...
> > > > Yes, but also nesting this in an outer loop is fairly trivial in the sense that it is a well known approach.
> > >
> > > If by well-known you mean that such nesting has been done with specific applications in mind, such as meta-learning the initialisation (MAML) or learning rate, then this plays precisely to the point of this section, namely that there is a commonality to all these specific approaches that fits under a fairly simple mathematical framework within which we can jointly reason about e.g. requirements. If by this you mean that there is some work providing a similar high level formalization, beyond Franceschi et al 2018 (which we compare against), which we should be mentioning here, we would be grateful if you could point us to this literature so that we can compare.
> > >
> > > > Page 4: there exist continuous hyperparam...
> > > > If they are not continuous then they should not even occur in this formalization so saying there exist... does not make much sense to me here
> > >
> > > Yes, this is just the antecedent of an entailment which, if it doesn’t hold, ensures that the consequent (the rest of the sentence) need not hold as a requirement. If this is still not clear, we can of course rephrase it.
> > >
> > > > \alpha \subset \varphi^{opt} implies that \varphi^{opt}  is a set from notation although we are treating it as a vector everywhere else
> > >
> > > You are right that this is an abuse of mathematical notation, but we frequently talk about a set of parameters being a subset (or having overlap) with others (e.g. when doing parameter sharing), and then treat them as vectors when defining the forward pass. We can describe this linguistically rather than mathematically if you think this is genuinely confusing. Please let us know your thoughts.
> > >
> > > > Page 4: All of section 2.4 seems somewhat trivial to me, but I guess that is highly subjective.
> > >
> > > Thank you for recognising the subjectivity of this value judgement. We would point out that any proof may seem trivial once presented, especially if it does not appear counter-intuitive to the reader (which evidently it does not to the reviewer). Our preference would be to leave it in the main body of the paper rather than an appendix, as some of the early readers of our paper found this section interesting or helpful (but that, too, is subjective).
> > >
> > > > Page 5: in the definition of stop operator perhaps use ⇔
> > >
> > > We’re not sure about this: the operator has to maintain the truth of joint constraints, so if either are false it does not satisfy the definition, therefore the “iff” doesn’t seem right. But perhaps we are misunderstanding your suggestion: if so, can you please clarify?
> > >
> > > > Page 5: Perhaps explicitly mention how your approach differs from a reverse mode differentiation of training or if it does not differ, say this.
> > >
> > > It’s a second order reverse mode. In fact, in unnamedlib, we basically rely upon pytorch’s autograd.grad function to build the reverse graph, and take higher order gradients over it. The library itself deals with ensuring that gradient paths which can exist, do. We will make a specific note of this at the end of the algorithm. Thanks for your suggestion.
> > >
> > > > Page 14: When talking about _S_GD (instead of just GD) perhaps mention something about non-existence of mini-batch randomness / being deterministic
> > >
> > > There’s no requirement of determinism if the data sampled from the data distribution is not a function of the model parameters. More broadly, the sort of meta-learning Gimli covers is about optimizing the actual optimization process used (rather than the ideal one), so if in practice we train our model through iterative methods with MC estimates of the batch gradient (e.g. through SGD), then that’s what we are meta-learning to improve. If this doesn’t answer your question,or you think there’s something specific we should clarify in the paper, please let us know an we will happily do it.

---

> > > > ### Comment · AnonReviewer2 · 2019-11-15
> > > > **Response to authors**
> > > >
> > > > Thank you for addressing my points so thoroughly.
> > > >
> > > > I am sorry that my review seems to be so subjective.
> > > > I will stick to my evaluation of a weak reject though.
> > > >
> > > > Considering the goal of "mathematical generalization" from my perspective it would have been desirable had the authors looked beyond the meta-learning literature.
> > > > From a mathematical perspective meta-learning is a bilevel optimization problem.
> > > > In order to find a general formal framework for meta-learning it may therefore make sense to relate to results that are well-known in the broad and established fields of bilevel optimization.
> > > > There is also a non-negligible amount of literature on conditions that need to be present in order to differentiate parametric optimization problems.
> > > >
> > > > A typical approach in bilevel optimization is to utilize the implicit function theorem in order to take derivatives of a lower level optimization problem with respect to hyper parameters.
> > > > That is generally not possible if the Hessian of the lower level problem at the local optimum is singular, as would typically be the case of machine learning.
> > > > But mentioning in the context of an attempted generalization of meta-learning seems relevant to me (to a typical bilevel optimization practitioner it would seem strange to differentiate the optimizer).
> > > >
> > > > If the goal is rather to present the software tools I think it would make sense to focus on software aspects.

---

> > > > > ### Author Response · Authors · 2019-11-15
> > > > > **Thanks for your response**
> > > > >
> > > > > Thank you for responding before the discussion period is up.
> > > > >
> > > > > We wish to assume good faith on the part of the reviewer, but it sounds like they are setting an impossible standard for publication. We have answered the concerns brought forth by the reviewer, by their own admission, and they have simply brought up new high-level reasons they cannot support publication (none of which is specific enough to answer). Specifically, we stress again that solving engineering problems as we do in section 4 to easily permit experiments of the sort we present in section 5 is:
> > > > > a) called for by the ICLR CFP
> > > > > b) a valuable contribution to the community
> > > > > c) not specifically argued against by the reviewer
> > > > >
> > > > > We leave it to the AC to judge whether a case for rejection stands since we have addressed the issues from the first round of comments, but we would have preferred if the reviewer had engaged with our points rather than find an excuse—any excuse—to reject the paper.

---

> ### Author Response · Authors · 2019-11-14
> **Paper update changelog**
>
> Dear Reviewer,
>
> We have made modifications to the manuscript based on our discussion. For your convenience, please find, below, a log of the changes made that pertain to our discussion. Please note that the paper has still been kept under the page limit as a result of these changes.
>
> Replaced “The proposed formalism provides tooling for analysing the provable requirements of the meta-optimization process, and allows for describing the process in general terms.” with “The proposed formalism allows us to describe the meta-optimization process in general terms and analyse its requirements.” as discussed.
>
> Removed “without loss of generality” where it would cause confusion,
> Replaced “estimate” with “approximate” to avoid confusion, as discussed.
>
> Left a footnote for requirement 2 to clarify the applicability of the requirement to stochastic optimizers.
>
> Added a clarification about the relation between the Gimli update algorithm and reverse-mode differentiation as the end of section 2.
>
> Thanks again for your excellent suggestions.

---

### Decision · Program_Chairs · 2019-12-19

**Decision:**

Reject

**Comment:**

The reviewers agree that the technical innovations presented in this paper are not great enough to justify acceptance.  The authors correctly point out to the reviewers that the ICLR CFP states that the topics of "implementation issues, parallelization, software platforms, hardware” are acceptable.  I would point out that most papers in these spaces describe *technical innovations* that enable improvements in "parallelization, software platforms, hardware" rather than implementations of these improvements.   However, it is certainly true that a software package is an acceptable (although less common) basis for a publication, provided is it sufficiently unique and impactful.  After pointing this out to the reviewers and collecting opinions, the reviewers do not feel the combined technical and software contributions of this paper are enough to justify acceptance.